# Tissue-resident Eomes+ NK cells are the major innate lymphoid cell population in human infant intestine

Adrian F. Sagebiel[1], Fenja Steinert[1], Sebastian Lunemann[1], Christian Körner[1], Renée R.C.E. Schreurs[2,3], Marcus Altfeld[1], Daniel Perez[4], Konrad Reinshagen[5] & Madeleine J. Bunders[1,2,3]

Innate lymphoid cells (ILC), including natural killer (NK) cells, are implicated in host-defense and tissue-growth. However, the composition and kinetics of NK cells in the intestine during the first year of life, when infants are first broadly exposed to exogenous antigens, are still unclear. Here we show that CD103+ NK cells are the major ILC population in the small intestines of infants. When compared to adult intestinal NK cells, infant intestinal NK cells exhibit a robust effector phenotype, characterized by Eomes, perforin and granzyme B expression, and superior degranulation capacity. Absolute intestinal NK cell numbers decrease gradually during the first year of life, coinciding with an influx of intestinal Eomes+ T cells; by contrast, epithelial NKp44+CD69+ NK cells with less cytotoxic capacity persist in adults. In conclusion, NK cells are abundant in infant intestines, where they can provide effector functions while Eomes+ T cell responses mature.

[1] Department of Virus Immunology, Heinrich Pette Institute, Leibniz Institute for Experimental Virology, 20251 Hamburg, Germany. [2] Department of Experimental Immunology, Amsterdam Infection & Immunity Institute, Amsterdam University Medical Center, University of Amsterdam, 1105 AZ Amsterdam, The Netherlands. [3] Department of Pediatrics, Emma Children's Hospital, Amsterdam University Medical Center, University of Amsterdam, 1105 AZ Amsterdam, The Netherlands. [4] Department of General, Visceral and Thoracic Surgery, University Medical Center Hamburg-Eppendorf, 20246 Hamburg, Germany. [5] Department of Pediatric Surgery, University Medical Center Hamburg-Eppendorf, 20246 Hamburg, Germany. Correspondence and requests for materials should be addressed to M.J.B. (email: madeleine.bunders@leibniz-hpi.de)

Natural killer (NK) cells are innate lymphocytes that lack antigen-specific T or B cell receptors[1–4] and contain cytotoxic granules, providing them with the capacity to kill virus-infected cells[5]. NK cells have been classified as part of an heterogeneous group of innate lymphoid cells (ILCs) and play an important role in host-defense and tissue repair[6–9]. NK cells have superior cytotoxic qualities compared to other ILCs[10,11], which are generally identified by expression of the IL-7 receptor-α chain (CD127) and referred to as innate counterparts of T helper cells (ILC1s, ILC2s and ILC3s)[12,13]. However, NK cells and ILC1s do share the capacity to produce tumor necrosis factor-α (TNF-α) and interferon gamma (IFN-γ)[10,11]. Recent studies show that ILCs in tissues are able to provide local protection against infections[6,14]. ILCs and NK cells are already present in tissues early in human development and can be found in fetal intestines[15–17]. However, challenges to obtain infant tissues after birth have resulted in a lack of studies investigating NK cells during this critical phase of human development. As a result most of our understanding of NK cell ontogeny in children is based on studies of NK cells in blood or tissues derived from older children[18–20]. Therefore, the composition and kinetics of NK cells in intestines during the first year of life, when infants are exposed to exogenous antigens and have a high susceptibility to viral infections, are still unclear[21].

Here we demonstrate that CD127⁻CD103⁺Eomes⁺ NK cells are the major ILC population in infant intestines during the first months of life, and that their absolute numbers decrease with age. Intestinal CD127⁺ ILCs are also present early in life, but to a lesser extent than NK cells. Infant intestinal NK cells exhibit a cytotoxic phenotype compared with adult intestinal NK cells, and have higher perforin and granzyme B expression combined with superior capacity to degranulate. The number of intestinal NK cells and CD127⁺ ILCs decreases as that of Eomes⁺ T cells increases. Meanwhile, the intestinal NK cell subset persisting into adulthood is characterized by high expression of NKp44. Thus, the first year of life features dynamic changes in the lymphocyte compartment, shifting from Eomes⁺ NK cells to Eomes⁺ T cells in human intestines.

## Results

### Expression of NK cell markers on infant intestinal NK cells.

ILCs are a heterogeneous population with different effector functions[6,9,10,12,17]. The lack of a hallmark lineage marker to distinguish NK cells from other ILC1s in tissues has led to conflicting results investigating ILCs[10,22–25]. Therefore, a detailed analysis of molecules expressed by NK cells, including CD16, CD56, CD127, CD7, KIR, CD94, NKp44, NKp46, NKp80, CD103, CD49a, and CD69 on viable CD45⁺CD3⁻CD14⁻CD19⁻ (lin⁻) lymphocytes was performed. Flow cytometric data of intestinal epithelium, lamina propria, or peripheral blood-derived viable CD45⁺lin⁻ lymphocytes was analyzed by dimensional reduction using viSNE algorithm[26]. The unsupervised approach of viSNE resulted in a tissue-depended clustering of viable CD45⁺lin⁻ lymphocytes, indicating phenotypic differences between intestinal epithelial, lamina propria, and peripheral blood-derived cells (Fig. 1a). After dimensional reduction, intestinal epithelium, lamina propria, and blood-derived cells were highlighted separately to discern surface expression of signature molecules on viable CD45⁺lin⁻ lymphocytes (Fig. 1b). CD56 was frequently expressed on infant epithelium, lamina propria, and blood-derived viable CD45⁺lin⁻ lymphocytes. Intestinal epithelial CD56⁺CD45⁺lin⁻ lymphocytes were detected in various cell clusters including CD127⁺CD45⁺lin⁻ and CD127⁻CD45⁺lin⁻ cells. The viSNE map of lamina propria-derived CD45⁺lin⁻ cells also showed CD127⁻ and

CD127⁺CD56⁺ hot spots. Thus, both CD127⁺ ILCs and NK cells expressed CD56 and therefore CD56 alone could not be employed to distinguish NK cells from non-cytotoxic ILC1s without CD127. The Fcγ receptor IIIa (CD16), frequently used to identify NK cells in blood, was indeed highly expressed by infant blood CD45⁺lin⁻ cells, whereas only a small fraction of intestinal epithelial and lamina propria-derived CD45⁺lin⁻ lymphocytes expressed CD16. KIR expression is considered a hallmark of NK cells within the ILC family[10]. A small cluster of KIR⁺CD56⁺lin⁻ cells was present in the intestinal epithelium and lamina propria of infants. CD127⁺ cells were absent within the CD45⁺lin⁻KIR⁺ cell cluster, verifying the exclusive expression of KIRs by NK cells. Next, we investigated whether other markers had a higher specificity to identify NK cells without requiring additional gating on CD127⁻ cells. The NK cell receptor CD94, which together with NKG2A binds to HLA-E[2], was largely co-localized with CD56 on CD45⁺lin⁻ cells and was not detected within CD127⁺ cell clusters. Gating on CD94⁺CD45⁺lin⁻ cells however, did not include all KIR⁺CD45⁺lin⁻ cell populations. A primary gating strategy based on CD94⁺ cells would therefore exclude a proportion of epithelial and lamina propria-derived KIR⁺ NK cells in infants. NKp80 is another surface marker that has been used to specifically identify NK cells[27]. Expression of NKp80 corresponded to a large extend to CD56 expression, however not all clusters identified by CD56 and CD94 were NKp80 positive, while at the same time there was overlap with CD127⁺ cells, thus not providing better sensitivity and specificity. The natural cytotoxicity receptors (NCRs) NKp46 and NKp44 have been described on both NK cells and ILCs[28]. NKp46 included a large number of CD45⁺lin⁻ cell clusters, including KIR⁺ cells, as well as CD127⁺lin⁻ cells from infant intestines, demonstrating that this NCR did not improve the selectivity of a gating strategy based on CD56 and CD127. The expression of NKp44 was less abundant than NKp46 on intestinal epithelium or lamina propria-derived CD45⁺lin⁻ cells. However, CD127⁺ and CD127⁻ clusters were distinguished amongst NKp44⁺ cells from infant epithelial and lamina propria tissues, eliminating NKp44 as a lineage marker for NK cells. Taken together, a gating strategy based on CD56⁺CD45⁺lin⁻ cells and excluding CD127⁺ cells (Fig. 2a) included all KIR⁺lin⁻ lymphocytes as well as NK cell populations identified by other markers, such as NKp80 and CD94 (Fig. 1b). Applying this gating strategy showed that CD16 expression by infant intestinal epithelial NK cells was low compared to infant peripheral blood cells, with only 16% (median, interquartile ranges (IQR) 11–21%) of infant epithelial NK cells expressing CD16 (Fig. 2a, c, Supplementary Fig. 1). Similar small frequencies of CD16⁺ NK cells were detected in adult small intestinal tissues. In conclusion, NK cells derived from infant blood and intestines differed significantly in their expression of hallmark surface markers. We furthermore established a gating strategy of CD127⁻CD56⁺CD45⁺lin⁻ cells that allowed for a comprehensive characterization of NK cells in infant intestines.

### NK cells form the major ILC population in infant intestines.

Using the gating strategy established above, the contribution of viable CD127⁻CD56⁺CD45⁺lin⁻ lymphocytes (hereafter called NK cells) to overall lymphocyte populations in infant intestines was quantified. In infants under 3 months of age, NK cells comprised a large proportion of epithelial (median 29%, IQR 17–35%) and lamina propria-derived (median 15%, IQR 8–26%) lymphocytes. In contrast, NK cells contributed only 4% (median, IQR 2–6%) to epithelial lymphocytes and 1% (median, IQR 1–2%) to lymphocytes in the lamina propria of adult intestines (Fig. 2d). Absolute NK cell numbers decreased after ~6 months of age (Fig. 2e, Supplementary Fig. 1). Intestines of infants under

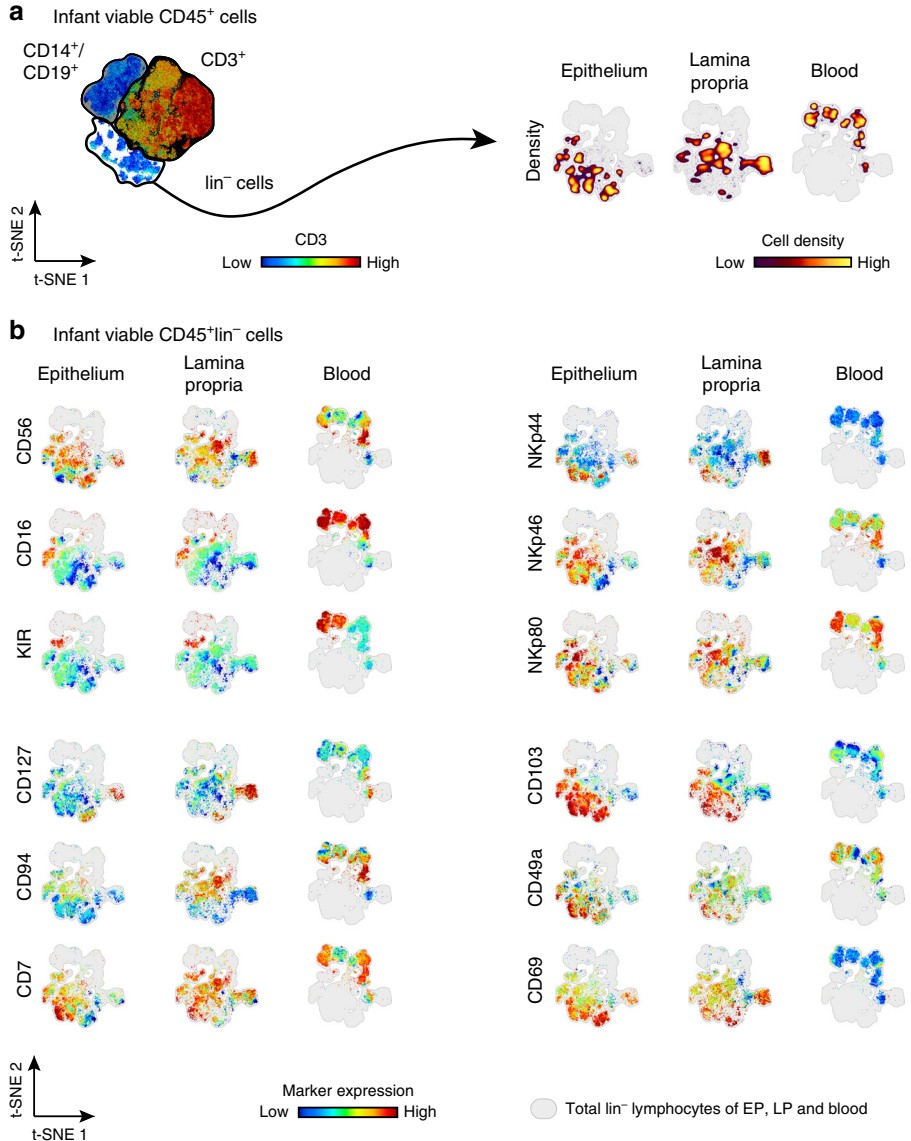

**Fig. 1** Hallmarks of ILCs by infant blood and intestinal CD45+lin− cells. **a** viSNE plot of flow cytometric analysis of viable CD45+ cells showing separate clustering by tissue of origin: epithelium (EP), lamina propria (LP), and blood (B). **b** Individual representation of hallmarks of ILCs (CD56, CD16, KIR, CD127, CD94, CD7, NKp44, NKp46, NKp80, CD103, CD49a, and CD69) using viSNE algorithm of flow cytometric data of lin−(CD14−CD19−CD3−) cells for EP, LP, and blood samples. Expression is shown by color coding in relative intensity. viSNE plots have been calculated from concatenated FCS files gated on viable CD45+ lymphocytes from epithelium, lamina propria, and blood of matched donors ($N = 4$, iterations = 7500 perplexity = 100, KL divergence = 2.15)

6 months of age contained 51,103 epithelial NK cells (median, IQR 44,198–69,504) per cm$^2$ compared to 2877 epithelial NK cells (median, IQR 2092–3819) in adult intestines. Absolute counts of lamina propria-derived NK cells/cm$^2$ decreased from 78,620 (median, IQR 39,274–112,743) in infants under 6 months of age to 4941 (median, IQR 3449−9762) in adults (Fig. 2e). Intestinal CD127+ ILCs showed a similar trend over age as NK cells. The highest absolute numbers and frequencies of epithelial and lamina propria-derived CD127+ ILCs were detected after birth and decreased thereafter (Supplementary Fig. 1). NK cell frequencies and absolute numbers remained higher compared to CD127+ ILCs in intestines during infancy (Fig. 2d, e and Supplementary Fig. 1). Taken together, intestines of infants after birth contained 17.8 times more epithelial ($p < 0.001$) and 15.9 times more lamina propria-derived ($p < 0.001$) NK cells, respectively, compared to adult intestines, indicating that NK cells are amongst the most numerous lymphocytes in intestines early in life.

**Infant intestinal NK cells have a tissue-residency phenotype**. Tissue-resident lymphocytes, including ILCs, allow for compartmentalization of immune responses adapted to local requirements[29–33]. CD69, CD103, and CD49a have been suggested to facilitate retention of lymphocytes in intestinal tissues[29–32]. Epithelial NK cells showed high expression of CD103 both in infant (median 90%, IQR 80–92%) and adult intestines (median 93%, IQR 78–97%) (Fig. 3a, b). Infant lamina propria-derived NK cells however exhibited a higher expression of CD49a compared to adult NK cells ($p = 0.003$). In line, viSNE plots showed absence of CD49a expression in those clusters with the highest density of adult lamina propria-derived NK cells (upper right location of the viSNE map) (Fig. 3a). In contrast, CD69 was expressed significantly lower on infant epithelial and lamina propria-derived NK cells compared to adult NK cells ($p < 0.001$ and $p < 0.001$, respectively) (Fig. 3b). Of note, CXCR6, identifying tissue-resident NK cells in the liver[34], was relatively rarely expressed

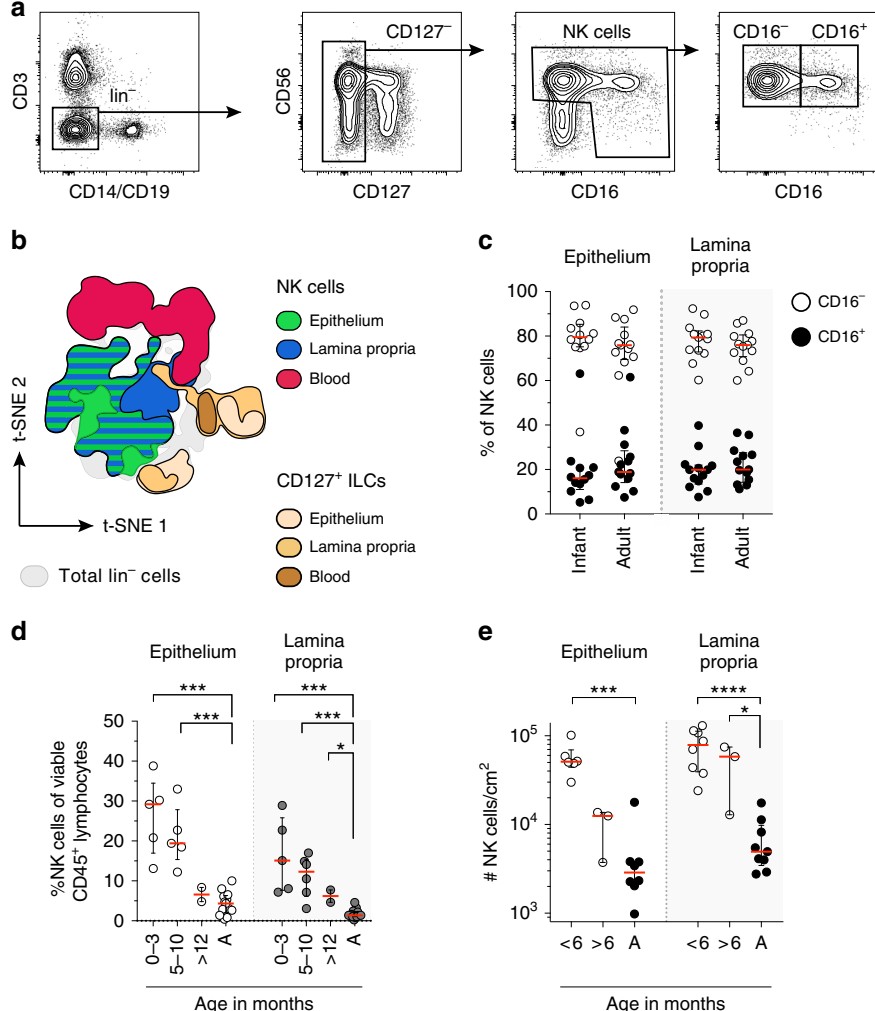

**Fig. 2** NK cells are abundant in infant intestines. **a** Flow cytometric plots showing the applied gating strategy for intestinal NK cells (viable CD56+CD127− CD45+lin− cells). FACS plots of a representative donor are shown. **b** viSNE plot of flow cytometric data showing gated NK cells and CD127+ ILCs from epithelium (EP), lamina propria (LP) and blood (B) within lin− populations (gray) (N = 4). **c** Frequencies of infant and adult CD16− (white circles) and CD16+ (black circles) NK cells from EP (N = 12) and LP (N = 13) tissues (infant samples), adult samples N = 13 (EP and LP). **d** NK cell frequencies of viable lymphocytes shown for different age groups in EP and LP tissues (infant samples N = 12 (EP) and N = 13 (LP), adult samples (A) N = 13 (EP and LP). **e** Absolute NK cell numbers per cm² for different age groups in EP and LP tissues (infant samples N = 9 (EP) and N = 11 (LP), adult samples (A) N = 8 (EP), and N = 9 (LP)). Median frequencies indicated by red lines. Error bars define interquartile ranges between 75th and 25th percentiles. Statistical comparisons are Mann-Whitney U comparisons. Asterisks represent the following p-values: *p < 0.05; ***p < 0.001; and ****p < 0.0001

on intestinal NK cells compared to CD103 or CD69 (Supplementary Fig. 2). Taken together, the majority of intestinal NK cells early in life were CD103+ phenotypic tissue-resident NK cells. Furthermore, CD69 expression was significantly lower on infant intestinal NK cells and increased with age.

**Infant intestinal NK cells have high NKG2A expression.** Expression of NKG2A, KIR, and CD57 has been suggested to track NK cell differentiation, with NKG2A levels decreasing while KIR and CD57 increase upon maturation[35]. viSNE analyses showed NKG2A-positive and KIR-positive clusters of infant epithelium and lamina propria-derived NK cells. Remarkably, NKG2A and KIR were co-expressed in a small cluster in the upper left area of the viSNE map, while CD57+ cells were scarce and did not form a separate cluster (Fig. 4a). NKG2A+ NK cells were 1.8-fold and 1.4-fold more frequent in epithelium (p = 0.003) and lamina propria (p = 0.002) of infant intestines, respectively, compared to adult intestines (Fig. 4b). Although

higher NKG2A expression by infant NK cells suggested a more immature phenotype, frequencies of KIR+ NK cells in infant intestines were also significantly higher compared to adult NK cells (Fig. 4b). Thus, expression of NKG2A and KIR significantly differed between adult and infant intestinal NK cells. Frequencies of CD57+ infant and adult intestinal NK cells were low (Supplementary Fig. 3). Next, we assessed expression of NKG2A and KIR on CD103+, CD49a+, or CD69+ NK cells derived from infant and adult intestines. NKG2A was significantly higher expressed on CD103+ and CD49a+ infant intestinal NK cells than on CD69+ NK cells (NKG2A expression on CD103+ vs. CD69+ epithelial NK cells: p = 0.002; NKG2A expression on infant CD49a+ vs. CD69+ lamina propria-derived NK cells: p = 0.008) (Fig. 4c, d). Moreover, intestinal epithelial and lamina propria-derived KIR+ NK cells were also significantly reduced in CD69+ compared to CD103+ NK cell populations (Fig. 4c, d). Overall, these data suggest that intestinal NK cells have a relatively immature phenotype; however, a KIR+ NK cell population is present, in particular in infant intestines. We next investigated

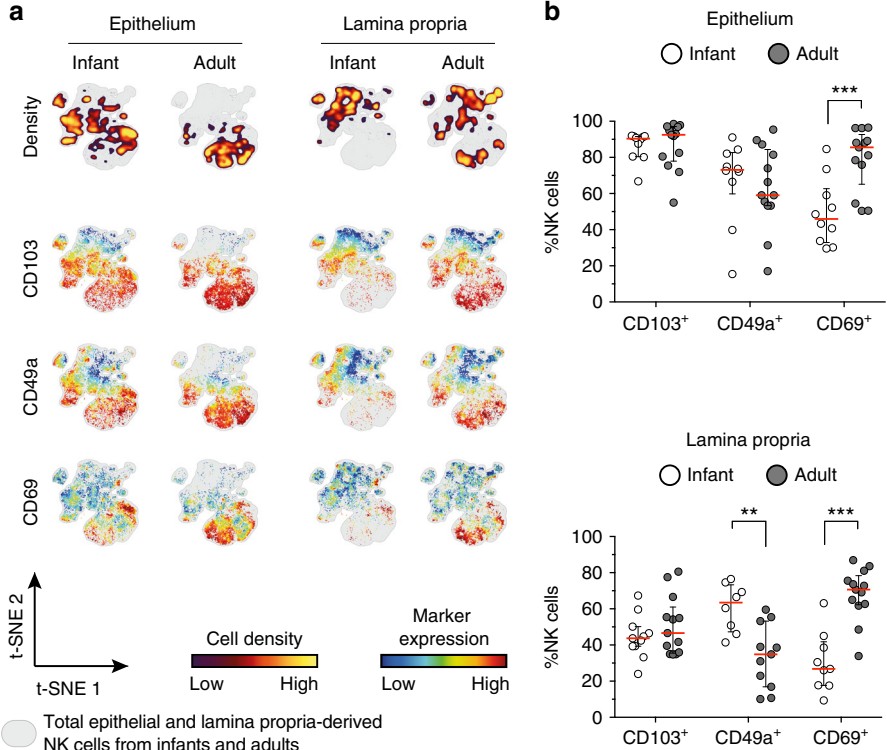

**Fig. 3** Intestinal NK cells express tissue-residency markers. **a** viSNE plots of combined flow cytometric data visualizing CD103, CD49a, and CD69 expression on infant and adult NK cells from epithelium (EP) and lamina propria (LP) intestinal tissues. Cell density of clusters is shown in first row. Expression patterns of tissue-residency markers CD103, CD49a, and CD69 are depicted by color coding in relative intensity in following plots. viSNE plots have been calculated from concatenated FCS files gated on NK cells (infant samples $N = 9$, adult samples $N = 6$, iterations = 7500 perplexity = 100, KL divergence = 2.29). **b** Frequencies of epithelial and lamina propria-derived CD103+, CD49a+, or CD69+ NK cells from infant (white circles) and adult intestines (dark circles) (EP infant samples ($N = 10$), EP adult samples ($N = 13$), LP infant samples (CD103 ($N = 11$), CD69 ($N = 9$), CD49a ($N = 8$), LP adult samples (CD103 ($N = 13$), CD69 ($N = 13$), CD49a ($N = 11$)). Median frequencies indicated by red lines. Error bars define interquartile ranges between 75th and 25th percentiles. Statistical comparisons are Mann-Whitney $U$ comparisons. Asterisks represent the following $p$-values: **$p < 0.01$ and ***$p < 0.001$

whether hallmark nuclear transcription factors for ILC and NK cell development might contribute to the differential NK cell phenotypes observed in infants and adults.

**Eomes+ NK cells are abundant in infant intestines**. The T-box transcription factors Eomesodermin (Eomes) and TBX21 (T-bet) are both essential for NK cell development. Whereas Eomes is more restricted to NK cell development, T-bet is also critical for induction and lineage-commitment of CD127+ ILCs[12]. Tissue-resident NK cells in livers have been described to be Eomes^high and T-bet^low[36,37]. Our analyses of infant intestines showed large clusters of epithelial Eomes+ NK cells in viSNE graphs, whereas adult epithelial Eomes+ NK cells were scarce (Fig. 5a). Quantitative analyses showed that 49% (median, IQR 38–69%) of infant epithelial NK cells and 86% (median, IQR 78–91%) of infant lamina propria-derived NK cells were Eomes+, 2.2-fold and 1.9-fold higher compared to adults ($p = 0.003$ and $p < 0.001$, respectively) (Fig. 5b). Infant intestinal NK cells were T-bet^low with only 12% (median, IQR 7–24%) of infant epithelial and 15% (median, IQR 10–26%) of lamina propria-derived NK cells expressing T-bet (Supplementary Fig. 4). The number of T-bet+ NK cells was further decreased in adult intestines. Furthermore, Eomes expression by intestinal NK cells differed between CD103+ and CD69+ cells, as CD103+ NK cells expressed significantly more Eomes compared to CD69+ NK cells (Fig. 5c, d). This was observed for infant intestinal epithelial NK cells ($p = 0.008$) as well as infant ($p = 0.02$) and adult ($p = 0.04$) lamina propria-derived NK cells (Fig. 5d). Taken together, a significantly

larger population of tissue-resident NK cells with high expression of Eomes was present in infant compared to adult intestines, with the highest Eomes expression detected amongst infant CD103+ NK cells, while Eomes expression among infant intestinal CD69+ NK cells was low.

**Superior functional capacity of infant intestinal NK cells**. Eomes+ NK cells are generally considered exemplar NK cells with the capacity to induce cytotoxicity upon release of granzyme B-containing and perforin-containing granules[10]. Approximately 54% (median, IQR 25–76%) of infant epithelial NK cells contained perforin, which was four times more ($p = 0.01$) than adult epithelial NK cells (median 12%, IQR 7–21%) (Fig. 6a, b). Furthermore, the majority of infant lamina propria-derived NK cells (median 68%, IQR 36–82%) contained perforin compared to 24% (median, IQR 14–43%) in adult intestines ($p = 0.008$). A similar trend was observed for granzyme B, with higher frequencies of granzyme B+ infant than adult intestinal NK cells (epithelium $p = 0.04$, lamina propria $p < 0.001$) (Fig. 6b). viSNE analyses of infant and adult donors further revealed co-expression of Eomes and KIR with perforin and granzyme B (Fig. 6a). Analogous to Eomes, infant epithelial CD103+ NK cells had significantly higher levels of perforin ($p = 0.02$) and granzyme B ($p = 0.008$) compared to CD69+ NK cells (Fig. 6c). Furthermore, infant epithelial CD49a+ NK cells contained significantly more granzyme B compared to CD69+ NK cells ($p = 0.008$). A similar pattern was observed when comparing lamina propria-derived CD103+ and CD69+ NK cells (Fig. 6c). The consistent lower expression of

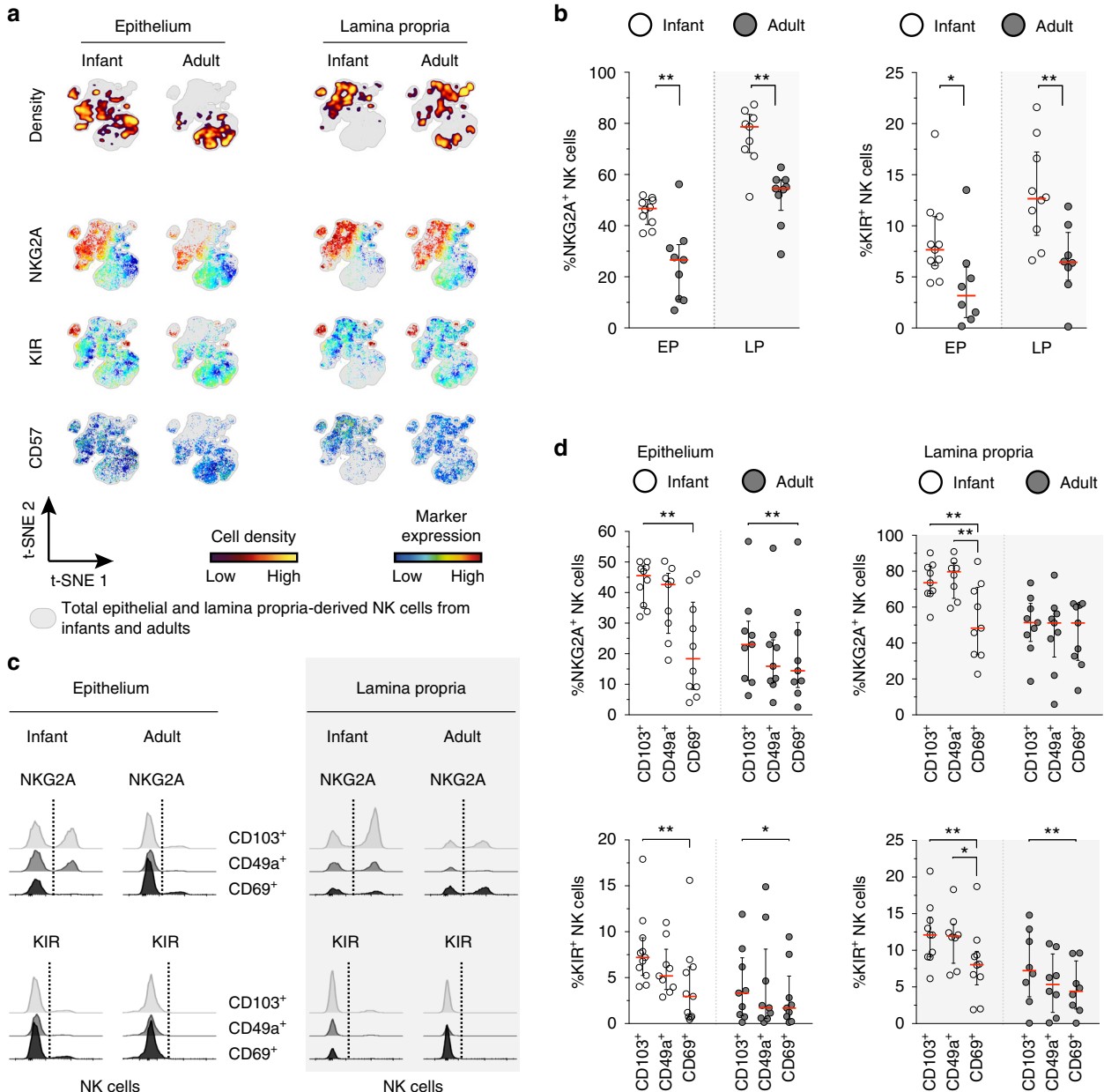

**Fig. 4** Infant intestinal NK cells have high NKG2A expression. **a** viSNE plots of combined flow cytometric data visualizing NKG2A, KIR, and CD57 expression by infant and adult epithelial (EP) and lamina propria-derived (LP) NK cells. Cell density of clusters is shown in first row. The expression of NKG2A, KIR, and CD57 is shown by color coding in relative intensity below. viSNE plots are calculated from concatenated FCS files gated on NK cells (infant samples $N = 9$, adult samples $N = 6$, iterations = 7500 perplexity = 100, KL divergence = 2.29). **b** Frequencies of epithelial and lamina propria-derived infant (white circles) and adult (dark circles) NKG2A+ NK cells and KIR+ NK cells (EP infant samples NKG2A ($N = 10$), KIR ($N = 11$), LP infant samples NKG2A ($N = 9$), KIR ($N = 10$), EP adult samples NKG2A ($N = 9$), KIR ($N = 8$), LP adult samples NKG2A ($N = 9$), KIR ($N = 8$)). **c** Histogram overlay of flow cytometric data showing NKG2A and KIR expression by CD103+ (gray), CD49a+ (dark gray), and CD69+ (black) NK cells from infant and adult intestines. **d** Frequencies of infant (white circles) and adult (dark circles) intestinal NKG2A+ NK cells and KIR+ NK cells within CD103+, CD49a+, or CD69+ populations. NKG2A expression by infant EP CD103+ ($N = 10$), CD69+ ($N = 10$), and CD49a+ ($N = 9$) NK cells, NKG2A expression by infant LP-derived CD103+ ($N = 9$), CD69+ ($N = 9$), and CD49a+ ($N = 8$) NK cells. NKG2A expression by adult EP and LP-derived NK cells ($N = 9$). KIR expression by infant EP CD103+ ($N = 11$), CD69+ ($N = 11$), and CD49a+ ($N = 9$) NK cells. KIR expression by infant LP-derived CD103+ ($N = 10$), CD69+ ($N = 10$), and CD49a+ ($N = 8$) NK cells. KIR expression by adult EP ($N = 9$) and LP-derived ($N = 8$) NK cells. Median frequencies indicated by red lines. Error bars define interquartile ranges between 75th and 25th percentiles. Statistical comparisons are Mann-Whitney $U$ comparisons (**b**) and Wilcoxon matched-pairs signed rank tests (**d**). Asterisks represent the following $p$-values: *$p < 0.05$ and **$p < 0.01$

granzyme B and perforin in infant CD69+ NK cells compared to CD103+ NK cells was not observed for adult epithelial NK cells, as almost all adult epithelial NK cells were double positive for CD103 and CD69 (Fig. 3b). However, adult CD103+ NK cells and CD69+ NK cells were present in the lamina propria and a

significant difference was detected between CD103+ and CD69+ NK cells (Fig. 6c). Taken together, these data show that infant intestinal NK cells, and in particular CD103+CD69- NK cells, contained more granzyme B and perforin than adult intestinal NK cells.

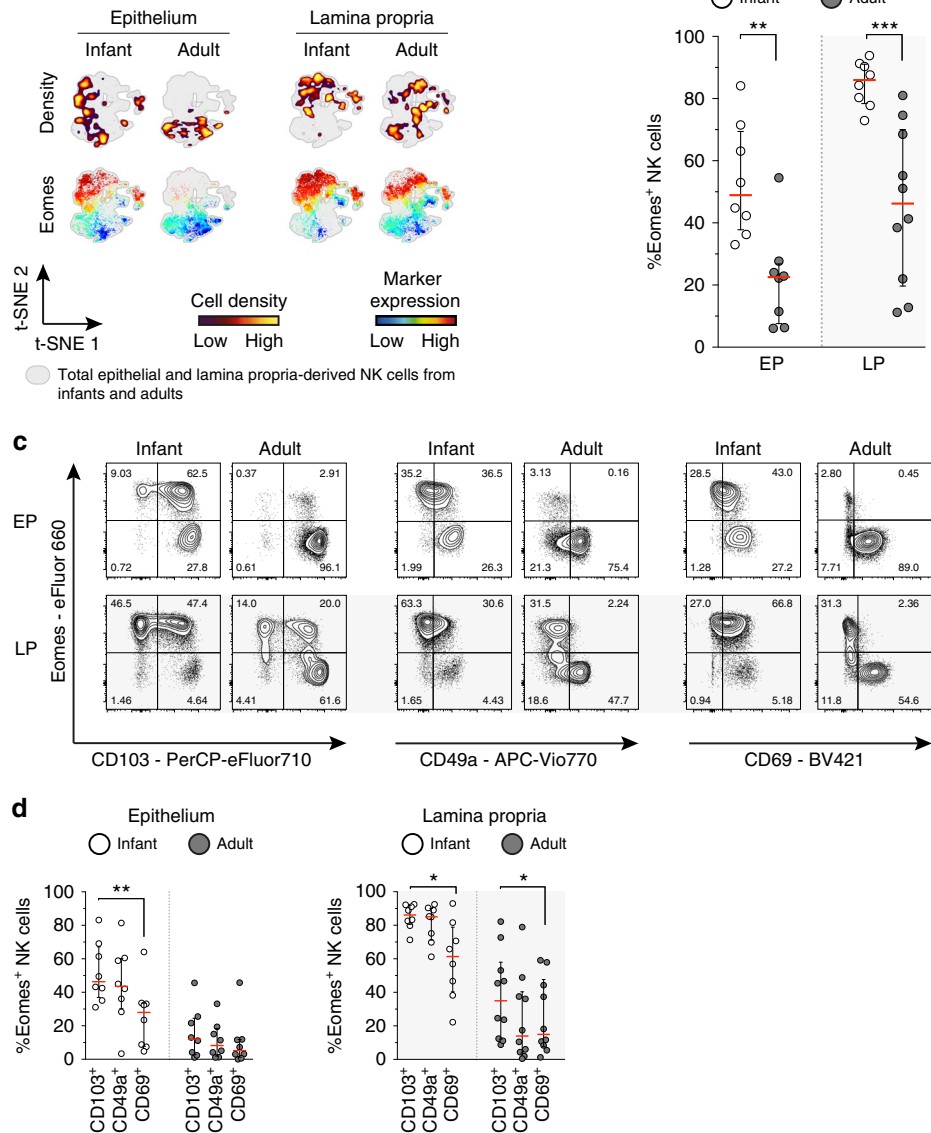

**Fig. 5** High expression of Eomes in infant intestinal NK cells. **a** viSNE plots of combined flow cytometric data visualizing Eomes expression of infant and adult NK cells from epithelium (EP) and lamina propria (LP) intestinal tissues. Cell density of clusters is shown in first row. Expression of Eomes is shown by color coding in relative intensity in second row. viSNE plots have been calculated from concatenated FCS files gated on NK cells (infant samples $N = 7$, adult samples $N = 5$, iterations = 7500 perplexity = 100, KL divergence = 2.62). **b** Frequencies of EP and LP-derived infant (white circles) and adult (dark circles) Eomes$^+$ NK cells. **c** Representative flow cytometric plots showing co-expression of Eomes and tissue-residency markers (CD103, CD49a, CD69) in NK cells from EP and LP of infant and adult intestines. **d** Frequencies of Eomes$^+$ NK cells within CD103$^+$, CD49a$^+$, or CD69$^+$ NK cell populations in EP and LP of infants (white circles) and adults (dark circles) (infant samples $N = 8$ (EP and LP), adult samples $N = 8$ (EP), and $N = 10$ (LP)). Median frequencies indicated by red lines. Error bars define interquartile ranges between 75th and 25th percentiles. Statistical comparisons are Mann-Whitney $U$ comparisons (**b**) and Wilcoxon matched-pairs signed rank tests (**d**). Asterisks represent the following p-values: *$p < 0.05$; **$p < 0.01$; and ***$p < 0.001$

The function of infant and adult intestinal NK cells was further assessed using degranulation and cytokine production assays. Due to low numbers of adult epithelial NK cells in intestines, these functional analyses were only performed using lamina propria-derived NK cells for comparisons. After isolation, lymphocytes were stimulated with phorbol-12-myristate-13-acetate (PMA) and ionomycin, K562 or 772.221 target cell lines. In infants, 77% (median, IQR 65–78%) of NK cells expressed CD107a upon stimulation with PMA and ionomycin, while only 54% (median, IQR 49–65%) of adult NK cells expressed CD107a ($p = 0.02$). Stimulation with PMA and ionomycin resulted in IFN-γ production by 43% (median, IQR 39–56%) of infant intestinal NK cells and 19% (median, IQR 18–54%) of adult

intestinal NK cells, whereas TNF-α was produced by 23% (median, IQR 8–50%) of infant and 9% (median, IQR 3–19%) of adult intestinal NK cells (Fig. 6d). After stimulation with K562 or 772.221 target cell lines, infant lamina propria-derived NK cells showed a trend towards enhanced degranulation compared to adult NK cells (Supplementary Fig. 5). Even though CD107a expression was relatively high on all NK cells, differential effects were observed between CD103$^+$ and CD69$^+$ NK cells (Fig. 6e). In particular adult intestinal CD69$^+$ NK cells showed a significant lower level of degranulation compared to CD103$^+$ NK cells ($p = 0.03$) (Fig. 6e), and a similar trend was observed for infant NK cells ($p = 0.06$). In conclusion, infant intestinal NK cells contained significantly more cytotoxic molecules and showed

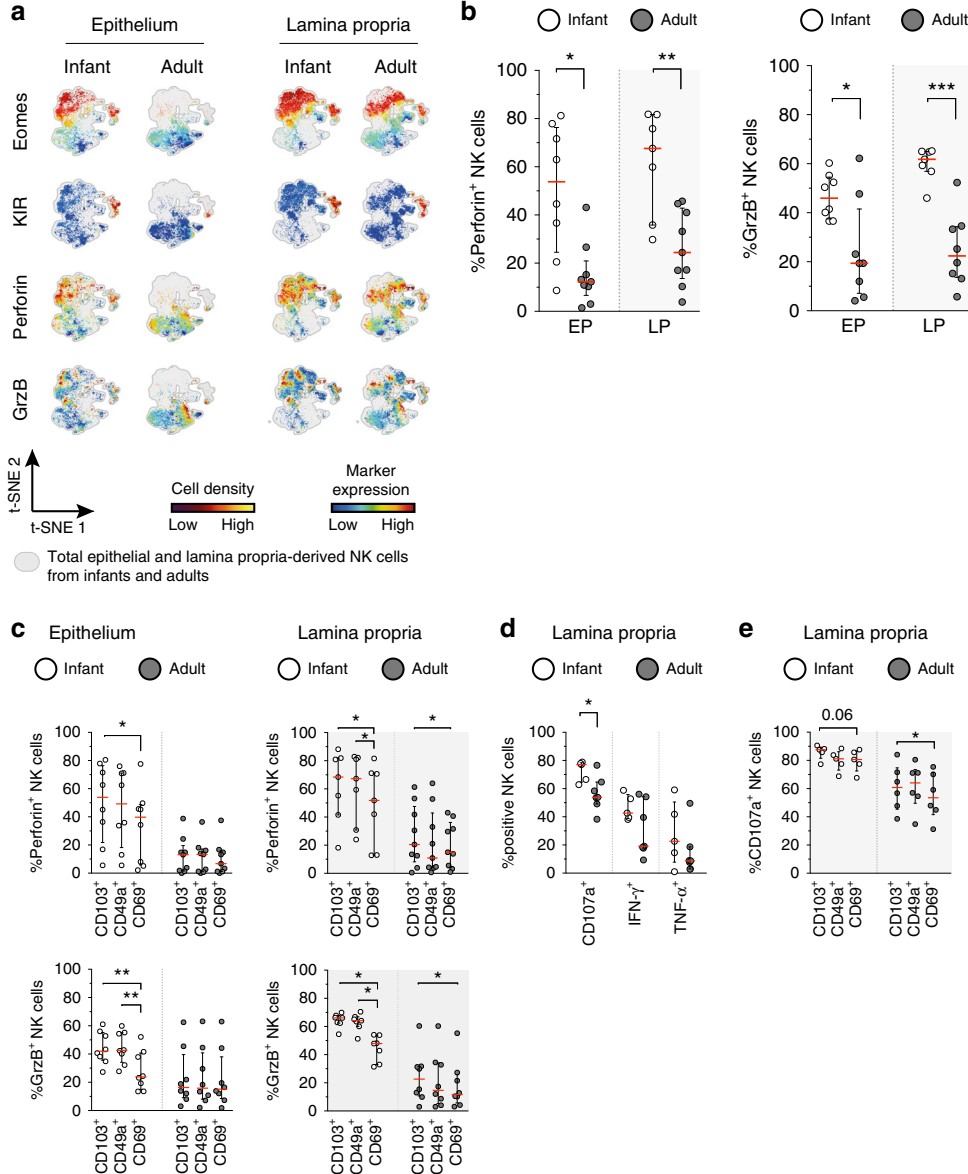

**Fig. 6** Infant intestinal NK cells contain high levels of cytotoxic granules. **a** viSNE plots of combined flow cytometric data visualizing Eomes, perforin, granzyme B, and KIR expression by epithelial (EP) and lamina propria-derived (LP) infant and adult NK cells. Expression of Eomes, KIR, perforin, and granzyme B (GrzB) is shown by color coding in relative intensity. viSNE plots have been calculated from concatenated FCS files gated on NK cells (infant samples $N = 7$, adult samples $N = 5$, iterations = 7500 perplexity = 100, KL divergence = 2.62). **b** Frequencies of perforin+ and GrzB+ NK cells in infants (white circles) and adults (dark circles) (EP infant samples ($N = 8$), LP infant samples ($N = 7$), EP adult samples perforin expression ($N = 9$), GrzB expression ($N = 8$), LP adult samples perforin expression ($N = 9$), and GrzB expression ($N = 8$)). **c** Frequencies of perforin+ and granzyme B+ cells within CD103+, CD49a+, or CD69+ NK cell populations in infant (white circles) and adult intestines (dark circles) (EP infant samples ($N = 8$), LP infant samples ($N = 7$), EP adult samples perforin expression ($N = 9$), and GrzB expression ($N = 8$), LP adult samples perforin expression ($N = 9$), and GrzB expression ($N = 8$)). **d** Frequencies of LP-derived CD107a+, IFN-γ+, and TNF-α+ NK cells in infant (white circles) and adult intestines (dark circles). Cells were stimulated for 6 h with phorbol 12-myristate 13-acetate (PMA) and ionomycin (infant samples $N = 5$, adult samples $N = 7$). **e** Frequencies of LP-derived CD107a+ cells within CD103+, CD49a+, or CD69+ NK cell populations in infant (white circles) and adult intestines (dark circles) after stimulation with PMA and ionomycin for 6 h (infant samples $N = 5$, adult samples $N = 6$). Median frequencies indicated by red lines. Error bars define interquartile ranges between 75th and 25th percentiles. Statistical comparisons are Mann-Whitney $U$ comparisons (**b**, **d**) and Wilcoxon matched-pairs signed rank tests (**c**, **e**). Asterisks represent the following p-values: *$p < 0.05$; **$p < 0.01$; and ***$p < 0.001$.

superior degranulation upon stimulation than adult intestinal NK cells.

**CD103+NKp44+CD127−lin− cells persist in adult intestines.** Our studies observed a highly dynamic intestinal NK cell compartment during childhood with an abundance of intestinal NK cells early in life followed by their rapid decrease. Recently, epithelial CD103+NKp44+ ILC1s were described in intestines of adults, which also exhibited certain features of NK cells, such as IFN-γ production and perforin expression[38,39]. We confirmed that epithelial NK cells in adult intestines expressed NKp44 (median 67%, IQR 54–72%) (Fig. 7a), whereas in infant intestines the percentage of epithelial NKp44+ NK cells was significantly

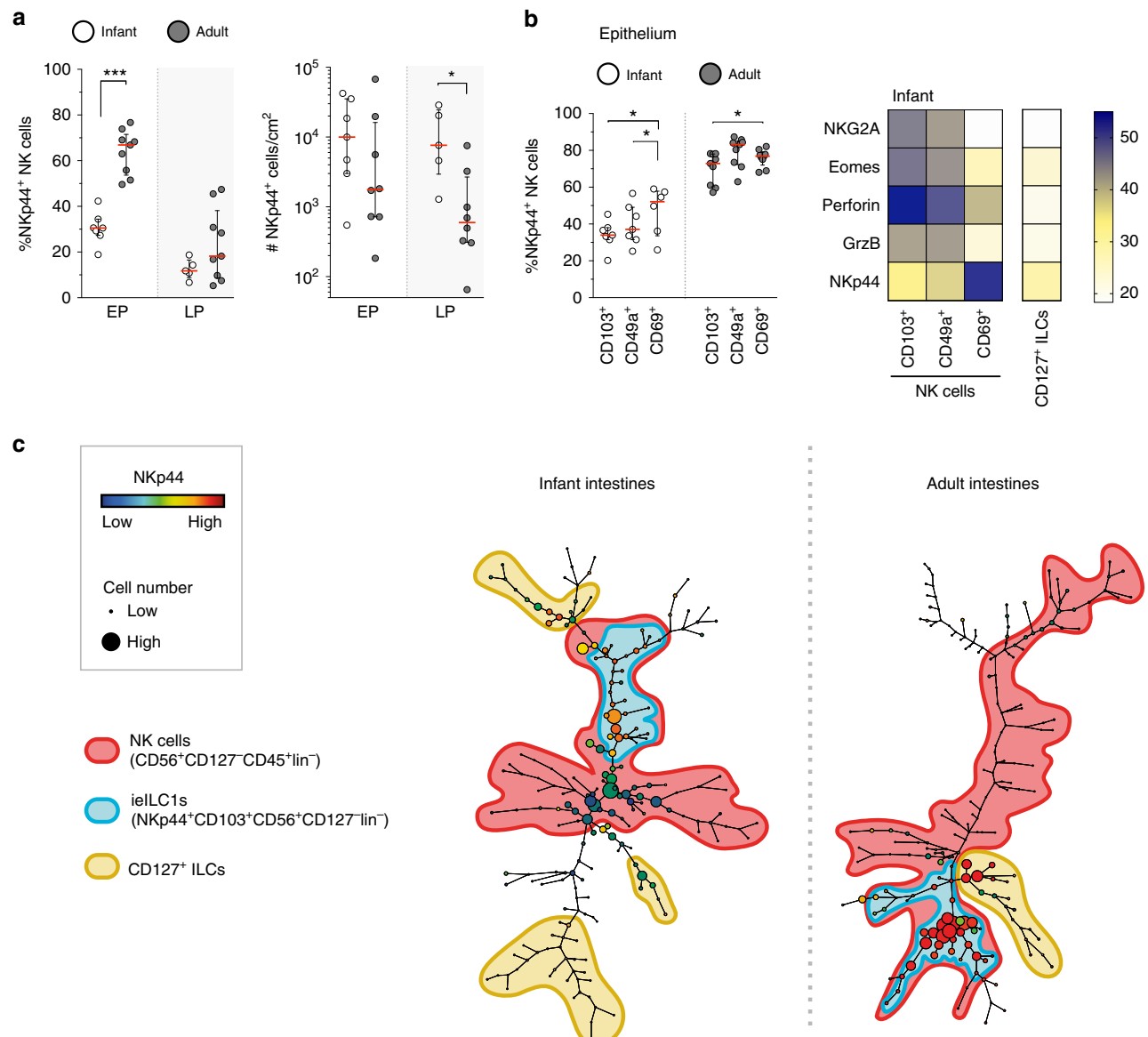

**Fig. 7** Epithelial CD103+NKp44+lin− cells are the major ILC population in adults. **a** Frequencies as well as absolute cell numbers per cm² of epithelial (EP) and lamina propria-derived (LP) NKp44+ NK cells in infant (white circles) and adult (dark circles) intestines (infant NK cell frequencies and absolute counts EP ($N = 7$) and LP ($N = 5$), adult NK cell frequencies ($N = 9$) and absolute counts ($N = 8$)). **b** Frequencies of NKp44+ cells within CD103+, CD49a+, or CD69+ NK cells in infants (white circles) and adults (dark circles). NKp44 expression by infant epithelial CD103+ ($N = 7$), CD49a+ ($N = 7$), and CD69+ ($N = 6$) NK cells. NKp44 expression by adult epithelial NK cell subsets ($N = 9$). Heatmap of median frequencies of NKG2A+, Eomes+, perforin+, granzyme B+, and NKp44+ cells within CD103+, CD49a+, or CD69+ NK cells as well as CD127+ ILCs in infant intestines. **c** SPADE tree of viable EP CD45+lin− lymphocytes. CD127+ ILCs (yellow outline), NK cells (red outline), including CD127−NKp44+CD103+lin− cells (blue outline) within NK cell population. Expression of NKp44 is shown by color coding in relative intensity. Node sizes represent the size of populations. SPADE analysis was computed by using the same signature parameters as in Fig. 1 (infant samples $N = 4$, adult samples $N = 5$, target number of nodes: 200, down sampled events target: 30%). Median frequencies indicated by red lines. Error bars define interquartile ranges between 75th and 25th percentiles. Statistical comparisons are Mann-Whitney $U$ comparisons (**a**) and Wilcoxon matched-pairs signed rank tests (**b**). Asterisks represent the following p-values: *$p < 0.05$ and ***$p < 0.001$

lower (median 31%, IQR 27−34% $p < 0.001$). Next, we assessed whether increased frequencies of NKp44+ NK cells in adult intestines were due to their selective expansion and observed that absolute numbers of epithelial NKp44+ NK cells were not higher in adults than in infants (Fig. 7a). Absolute numbers of lamina propria-derived NKp44+ NK cells were in fact higher in infant compared to adult intestines ($p = 0.01$). Thus, NKp44 expression identifies an NK cell subset persisting in intestines from infancy to adulthood. A heatmap analysis of investigated NK cell markers illustrated this further by identifying three clusters of infant intestinal lin− cells. The first cell cluster was defined by high

expression of NKG2A, perforin, granzyme B, and Eomes, representing NK cells (Fig. 7b). A second cell cluster was identified by high expression of NKp44 and CD69, representing previously described ieILC1s[38]. The third cluster comprised CD127+ ILCs. Intestinal CD127+ ILCs were CD69+ with low expression of CD103 and CD49a, in particular adult lamina propria-derived CD127+ ILCs. Infant intestinal CD127+ ILCs differed from NK cells, as only low frequencies of CD127+ ILCs expressed either NKG2A or KIR (Supplementary Fig. 6). Eomes was detected in only 8% (median, IQR 1−12%) of infant and 12% (median, IQR 1−49%) of adult epithelial CD127+ ILCs. Intestinal CD127+ ILCs

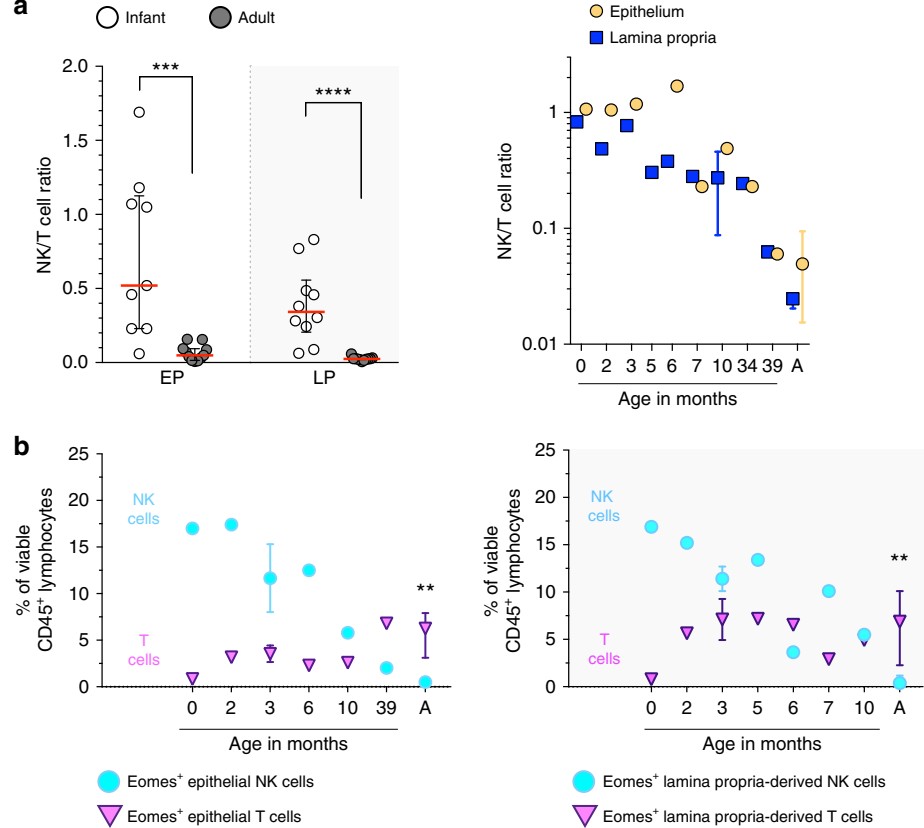

**Fig. 8** Increase of intestinal Eomes$^+$ T cells in the first year of life. **a** NK-T cell ratios of infant (white circles) and adult intestines (dark circles, A) in epithelium (EP, yellow circles) and lamina propria (LP, blue squares) (infant EP samples ($N = 9$) and LP samples ($N = 10$), adult EP and LP samples ($N = 11$). **b** Frequencies of epithelium and lamina propria-derived Eomes$^+$ NK (turquoise circles) and Eomes$^+$ T cells (violet triangles) within the total lymphocyte pool at different ages (infant EP and LP samples ($N = 7$) and adult EP and LP samples ($N = 10$)). Median frequencies indicated by red lines. Error bars define interquartile ranges between 75th and 25th percentiles. Statistical comparisons are Mann-Whitney $U$ comparisons. Asterisks represent the following $p$-values: **$p < 0.01$; ***$p < 0.001$; and ****$p < 0.0001$

furthermore exhibited a non-cytotoxic phenotype, as perforin and granzyme B expressions were low in both infant and adult intestinal CD127$^+$ ILCs (Supplementary Fig. 6). NKp44 was expressed by 52% (median, IQR 44–66%) of infant and 50% (median, IQR 31–69%) of adult CD127$^+$ ILCs. The phenotype of CD127$^+$ ILCs more closely resembled the cluster of NKp44$^+$ CD69$^+$ NK cells in infant and adult intestines. Although functional responses of infant CD127$^+$ ILCs were relatively heterogeneous upon PMA and ionomycin stimulation, frequencies CD107a$^+$CD127$^+$ ILCs were relatively low, as was IFN-γ and TNF-α production (Supplementary Fig. 6). Taken together, innate lymphocyte populations in infant intestines consisted of a large population of prototypic CD127$^-$ NK cells and a smaller population of CD127$^+$ ILCs, which both declined during childhood, while a population of NKp44$^+$CD69$^+$ ieILC1s persisted.

To further illustrate dynamic changes of these ILC populations, we performed a spanning-tree progression analysis of density-normalized events (SPADE)[40] of multi-parameter flow cytometry data, based on CD16, CD56, CD127, CD7, KIR, CD94, NKp44, NKp46, NKp80, CD103, CD49a, and CD69 by viable CD45$^+$ lin$^-$ lymphocytes. SPADE allowed to determine phenotypic hierarchies of different populations of lin$^-$ lymphocytes in infant and adult intestines. The size of a respective circle indicates the size of the population, whereas distances between circles in the tree reveal phenotypic similarity of different populations. The SPADE tree showed that NK cells constituted the largest population of infant epithelial lin$^-$ lymphocytes (Fig. 7c). In contrast, in the SPADE tree of adult epithelial lin$^-$ lymphocytes, NKp44$^+$CD103$^+$

ieILC1s were most abundant. Together these results show that a population of intestinal NKp44$^+$lin$^-$ cells remained relatively stable over life; however, its contribution to the overall innate lymphocyte population in intestines increased over age due to a decrease of NKp44$^-$ NK cells during infancy.

**NK cell decline coincides with Eomes$^+$ T cell accumulation.** Infants are well-known for immaturity of their adaptive immune system, in particular cytotoxic CD8$^+$ T cells[41]. In line with this, we have previously shown that at birth the majority of intestinal T cells are CD4$^+$ T cells whereas cytotoxic CD8$^+$ T cells are scarce[42]. However, the CD4/CD8 T cell ratio changes over age, and in adults an abundance of epithelial CD103$^+$CD69$^+$Eomes$^+$ CD8$^+$ T cells has been shown to provide protection against intracellular pathogens[43,44]. We therefore examined ratios between NK cells and T cells in intestinal samples over age. Epithelial NK/T ratios rapidly declined from 0.52 (median, IQR 0.23–1.1) in infant intestines to 0.05 (median, IQR 0.02–0.09) in adult intestines ($p < 0.001$) (Fig. 8a). A similar trend was observed for NK/T cell ratios in intestinal lamina propria ($p < 0.001$) (Fig. 8a). These findings indicated a transition from an evenly balanced NK/T cell immune system in infant intestines to a T cell-dominated immune system in adult intestines. Frequencies of Eomes$^+$ T cells were determined to investigate whether a maturation of cytotoxic T cell responses coincided with changes in NK cell populations. At birth, intestinal epithelial and lamina propria-derived Eomes$^+$ T cells were rare. After birth,

Eomes+ T cells increased, with highest frequencies of Eomes+ T cells detected in adult intestines. Intestinal epithelial Eomes+ T cells contributed 6% (median, IQR 3–8%) to the total epithelial lymphocyte population in adults and lamina propria-derived Eomes+ T cells contributed 7% (median, IQR 2–10%) to the total lamina propria-derived lymphocyte population in adults (Fig. 8b). Taken together, the first year of life was characterized by rapid changes in intestinal NK cell and T cell compartments, with Eomes+ NK cells declining and Eomes+ T cells increasing.

## Discussion

An increasing number of studies have shown that ILCs, including NK cells, are critical in host defense and can mediate tissue (re) modeling[6,8–10,12,13,38,45]. These studies primarily focused on fetal tissues or older children and adults[6,8–10,12,13,17,20,38,45]. However, studies investigating NK cells in infant intestines in the first year of life, when microbial colonization as well as rapid growth of mucosal tissues take place in concert, are lacking. Here, we demonstrate that NK cells represent the major innate lymphocyte population in small intestines of infants, whereas fewer NK cells persist in adult intestines. Infant intestinal NK cells demonstrated a strong effector phenotype characterized by Eomes, perforin, and granzyme B-expression compared to adult intestinal NK cells. NK cells decreased in infant intestines during the first year of life, concurring with an increase of Eomes+ T cells. A population of NKp44+CD103+CD69+ NK cells persisted in adult intestines, corresponding to previously described ieILC1s[38]. In conclusion, numerous prototypic NK cells populated small intestines in infancy and exhibited a cytotoxic effector profile, while adaptive immune responses were still immature.

Characterization of ILCs and NK cells is challenging in particularly in tissues due to lack of hallmark lineage markers for tissue-derived ILCs and NK cells[10,22–25]. Our analyses underlined the heterogeneity of ILCs and possible selection bias when for example only NKp80 or CD94 were included to identify NK cells. The well-documented plasticity of ILCs, varying origins of intestinal tissues included in previous studies and diverse pathologies may have further contributed to conflicting results reported on tissue-derived NK cells and ILCs[10,22–25,38,39]. Nonetheless, in general, non-cytotoxic ILCs are identified as CD127+lin− cells, while NK cells are CD127−Eomes+lin− cells[10,46] and contain cytotoxic granules with perforin and granzyme B. Here we employed a comprehensive approach to characterize ILCs and developed a gating strategy that allowed to identify infant intestinal NK cells. Our findings show that a gating strategy based on CD56+CD127−lin− lymphocytes identified a large population of bona fide CD127−Eomes+ NK cells in infant intestines. Infant intestinal NK cells expressed integrin αE (CD103), indicating that these cells are prompted to remain tissue-resident[33]. In addition to NK cells, infant intestines contained CD127+lin− ILC and NKp44+CD69+CD56+CD127−NK cell populations, which had lower Eomes and perforin expression. The NKp44+CD69+CD56+CD127−NK cell population persisted in adult intestines, as previously described[38]. Thus NK cells in infant intestines have a distinct phenotype compared to adult intestines.

Previous studies have described NK cells and non-cytotoxic ILCs in fetal tissues[16,17]. Although the role of ILCs as lymphoid tissue-inducers is well defined[47], the physiological role of intestinal NK cells prior to birth remains unknown. We detected the largest numbers of intestinal NK cells from birth to 6 months of age, suggesting that local NK cell populations are established early during human development. In line with previous data from fetal liver and lung tissues[48], infant intestinal NK cells had a relatively immature phenotype, but were equipped with cytolytic granules

and had superior degranulation capacity compared to adult NK cells. The decrease of infant intestinal NK cell and CD127+ ILC numbers coincided with an influx of Eomes+ T cells in infant intestines. The intestinal epithelium of adults contains large numbers of tissue-resident CD8+ T cells, which provide local antiviral immunity[43,49]. Our study shows that during human immune ontogeny, NK cells and CD127+ ILCs preceded adaptive cytotoxic CD8+ T cell responses. Infant intestinal NK cells were equipped with cytotoxic granules providing early innate effector responses, whilst CD8+ T cell responses still developed. Changes occurring in intestinal lymphocyte populations in the first year of life are reminiscent of the contraction of the NK cell pool during viral infection, where NK cells constitute early antiviral responses followed by their rapid decrease upon induction of Eomes+ CD8+ T cells. A competitive disadvantage of NK cells compared to CD8+ T cells for cytokines, such as IL-2 has been suggested as an underlying mechanism[50–53]. A similar developmental program appears to take place during immune ontogeny in infancy, suggesting a general pathway of immune constitution in different settings. Due to limitations in obtaining longitudinal human intestinal samples from birth to adulthood, the results presented here are derived from cross-sectional data. A causal sequence of events is therefore not possible. However, analyses of human intestines at different ages demonstrated a consistent pattern of maturation of the intestinal lymphocyte compartment. The exact mechanisms underlying dynamic modifications in the lymphocyte compartment in intestines, such as diet and maturation of the microbiome, need to be determined in future studies.

Our data further shows that the decline of infant intestinal NK cells and CD127+ ILCs primarily affected NKp44− populations, whereas NKp44+CD103+CD69+ NK cells persisted in adult intestines. NKp44+CD103+CD69+ NK cells have been previously described in adult intestines and, although classified as ILC1s, share similarities with NK cells[38]. In the SPADE tree of adult epithelial lin− lymphocytes, CD127+lin− cell populations and NKp44+CD127−NK cell/ieILC1 populations clustered closely together, indicating similarity. Future studies investigating these cell populations need to determine their hallmark cytokines. Next to a decline of NKp44− NK cells we furthermore observed a decrease of CD127+ ILCs in infant intestines. This suggests that the decrease of infant intestinal NK cells was not due to ILC plasticity[17,25] or transition of CD127− NK cells into CD127+ ILCs. As mentioned above, competition for cytokines is an important factor modulating the composition of lymphocyte populations in different anatomical locations[50–53]. Tissue-resident CD8+ T cells also express CD127[44,54]. Thus, upon influx of CD8+ T cells IL-2 and IL-7 are likely depleted, reducing survival signals for both NK cells and CD127+ ILCs. In contrast NKp44+CD103+ ieILC1s seemed relatively resistant to this influx of CD8+ T cells. Fuchs et al. showed that NKp44+CD103+ ieILC1s have a higher expression of CD122 (IL2-Rβ) compared to NKp44−lin− lymphocytes and ILC3s, potentially allowing NKp44+CD103+ ieILC1s an enhanced usage of IL-2 for survival. Taking together, the first year of life is characterized by highly dynamic changes within intestinal lymphocyte populations.

In conclusion, our study demonstrates that intestinal NK cells represent the major cytotoxic lymphocyte population early in human intestinal development. During the first year of life intestinal NK cells as well as CD127+ ILCs declined upon colonization of intestinal tissues with Eomes+ T cells, with a population of NKp44+CD103+CD69+ ieILC1s persisting in adult intestines. The first year of life is characterized by exposure to large numbers of microbes and changes in dietary intake. These factors have been shown in animal models to impact intestinal immune cell populations with repercussions for development of diseases later in life[55–58]. Our findings show that also in humans

significant modifications of innate lymphocyte populations take place during this dynamic phase of development, which may underlie specific pathologies observed in infancy.

## Methods

**Human tissue sample collection.** Human tissues were collected after donors (adults) or their guardians (infants) provided written informed consent. Pediatric small intestinal tissues (median age 5.5 months, IQR 2.3–9.3, $N = 16$) were obtained upon surgery to correct gastrointestinal congenital abnormalities and reconstruction of ileostomy. Adult small intestinal samples (median age 57 years, IQR 42.3–65, $N = 13$) were collected upon ileostomy reconstructions. None of the donors suffered from inflammatory diseases. Tissues were obtained with approval of the ethics committee of the Medical Association of the Freie Hansestadt Hamburg (Ärztekammer Hamburg).

**Lymphocyte isolation from human blood and intestinal tissues.** Tissue and blood samples were transported at 4 °C and processed in the laboratory within 6 h after surgery. The mononuclear cell fraction was isolated from blood using a density gradient. Blood was diluted 1:1 with Hank's balanced salt solution (HBSS; Sigma-Aldrich), then layered on top of BIOCOLL (Biochrom GmbH) and centrifuged. The mononuclear cell fraction was aspirated and washed with phosphate buffered saline (PBS). Intestinal tissues obtained at surgery were first washed with PBS to remove feces and blood. The muscular layer was removed. The sizes of intestinal tissues were documented after removal of the muscular layer. Intestinal tissues were next cut into $0.5 \times 0.5$ cm segments and incubated for $2 \times 20$ min, at 37 °C with Iscove's modified Dulbecco's medium (IMDM; Thermo Fisher Scientific) supplemented with 5 mM ethylenediaminetetraacetic acid (EDTA; Sigma-Aldrich), 2 mM 1,4-dithiothreitol (DTT; Carl Roth GmbH+Co. KG) and 1% fetal bovine serum (FBS; Biochrom GmbH) to detach the epithelial layer. Supernatant was filtered through a 70 μm cell strainer to obtain a single cell solution. Epithelial lymphocytes were isolated by density gradient centrifugation using BIOCOLL (Biochrom GmbH). The remaining intestinal tissue was minced and digested for $2 \times 30$ min at 37 °C with IMDM (Thermo Fisher Scientific) supplemented with 1 mg/ml Collagenase D (Sigma-Aldrich), 1% FBS (Biochrom GmbH) and 1000 U/ml DNAse I (STEMCELL Technologies). Supernatant containing cells was filtered through a 70 μm strainer to obtain a single cell solution. Lamina propria lymphocytes were isolated from single cell suspensions using a Percoll gradient (VWR International); standard isotonic Percoll solution (SIP) was prepared by supplementing 100% Percoll with 10% 10X PBS, using an additional 1X PBS which resulted in 60% SIP solution. After isolation, the numbers of viable cells were counted using Trypan blue.

**Flow cytometric analyses.** Isolated lymphocytes from human infant and adult small intestines were analyzed using 18-parameter flow cytometry. For surface staining, cells were incubated in PBS with the appropriate monoclonal antibodies and Zombie Aqua™ for 30 min at 4 °C, washed and fixed with 1X stabilizing fixative (BD Biosciences).

The following monoclonal antibodies (all anti-human) were used for surface staining (clone, catalog number, dilution): CD3-BUV395 (UCHT1, 563546, 1:80), CD56-BV786 (NCAM16.2, 564058, 1:100), CD16-BUV737 (3G8, 564434, 1:80), CD57-BV605 (NK-1, 563895, 1:160) from BD Bioscience. CD45-BV711 (HI30, 304049, 1:100), CD45-Alexa Fluor 700 (2D1, 368514, 1:80), CD14-PE-Cy7 (M5E2, 301814, 1:100), CD14-BV510 (M5E2, 301842, 1:100), CD19-PE-Cy7 (SJ25C1, 363011, 1:100), CD19-BV510 (HIB19, 302242, 1:100), CD127-PE-Dazzle594 (A019D5, 351336, 1:100), CD103-PE-Cy7 (Ber-ACT8, 350212, 1:100), CD69-BV711 (FN50, 310944, 1:80), CD69-BV605 (FN50, 310938, 1:50), CD69-BV421 (FN50, 310930, 1:200), NKp46/CD335-BV421 (9E2, 331913, 1:40), NKp44/CD336-PE (P44-8, 352107, 1:40), CXCR6/CD186-PE-Cy7 (K041E5, 356012, 1:20), CD107a-BV421 (H4A3, 328626, 1:40) from BioLegend. CD103-PerCP-eFluor710 (Ber-ACT8, 46-1037-42, 1:40), CD7-APC-eFluor780 (eBio124-1D1, 47-0079-41, 1:80) from eBioscience. CD94-FITC (REA113, 130-098-975, 1:40), NKG2A/CD159a-APC (REA110, 130-098-809, 1:40), NKp44/CD336-PE-Vio770 (2.29, 130-104-195, 1:40), KIR2D/CD158a-PE (NKVFS1, 130-092-688, 1:160), KIR3DL1/DL2/CD158e/k-PE (REA970, 130-095-205, 1:80), CD49a-APC-Vio770 (TS2/7, 130-101-406, 1:40) from Miltenyi Biotec. hNKp80-APC (239127, FAB1900A, 1:40) from R&D Systems.

For intracellular staining, surface-stained cells were washed, then incubated with 1X Cytofix reagent (eBioscience), washed again and incubated with 1X Cytoperm reagent (eBioscience) and the appropriate monoclonal antibodies for 30 min at 4 °C. The following monoclonal antibodies were used for intracellular staining: T-bet-BV711 (4B10, 644819, 1:40), perforin-PerCP-Cy5.5 (d9G, 308114, 1:40), IFN-γ-FITC (B27, 506504, 1:40), TNF-α-BV605 (MAb11, 502936, 1:40), granzyme B-FITC (GB11, 515403, 1:40) from BioLegend. Eomes-eFluor 660 (WD1928, 50-4877-42, 1:40) from eBioscience. Zombie Aqua™ Fixable Viability Kit (BioLegend) was used to determine cell viability. Stained cells were analyzed using a BD LSRFortessa cell analyzer (BD Biosciences) within 24 h and data was analyzed using FlowJo software v10 (TreeStar, Ashland, Oregon, USA).

**NK cell degranulation and cytokine production assay.** Intestinal isolated monocuclear cells were resuspended in IMDM with 10% FBS, and either left unstimulated, stimulated with phorbol 12-myristate 13-acetate (PMA) (Sigma-Aldrich), and ionomycin (SantaCruz Biotechnology), K562 or 772.221 target cells (Effector/Target cell-ratio: 1:5), in the presence of Brefeldin A (Sigma Aldrich), Monensin (BD), and anti-CD107a-BV421 (BioLegend) for 6 h at 37 °C and 5% $CO_2$. The K562 cell line was obtained from the Leibniz institute DSMZ-German Collection of microorganisms and cell cultures. The 772.221 cell line was obtained from America Type Culture Collection (ATCC). Cells were intracellularly stained for cytokines and analyzed as described above. Values of response parameters (CD107a, IFN-γ, and TNF-α positive cells) from stimulated conditions were corrected with corresponding values of unstimulated conditions:

$$\text{Relative value} = \frac{(\text{percentage}_{\text{stim.cells}} - \text{percentage}_{\text{unstim.cells}}) * 100}{(100 - \text{percentage}_{\text{unstim.cells}})}$$

**Statistical analyses.** The GraphPad Prism 7 (GraphPad Software, San Diego, CA) was used to analyze data and to perform statistical analyses. Statistical significance of differences was assessed using non-parametric Mann-Whitney U tests, or Wilcoxon matched-pairs signed rank test, for paired samples. Median frequencies and IQR are given in figures and text unless otherwise stated. Values of $p < 0.05$ were considered significant. NK cell populations were analyzed by dimensional reduction using Barnes-Hut $t$-distributed stochastic neighbor embedding (bht-SNE or viSNE) algorithm[26] and SPADE clustering[40] provided by the Cytobank platform (Cytobank Inc., Santa Clara, CA).

## Data availability

Data used in this study have been collected in a clinical study and are subject to regulations of the Ethics Committee of the Ärztekammer Hamburg that approved these studies. Participant's written consent has been provided for data generation and handling according to approved protocols. Data storage is performed by the Heinrich Pette Institute. A Reporting Summary for this article is available as a Supplementary Information file. Data are available upon request from the corresponding author and can be shared after confirming that data will be used within the scope of the originally provided informed consent.

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

## Acknowledgements

We would like to thank all donors and their parents for participation in this study. Further we thank the colleagues from the Department of Pediatric Surgery, as well as Department of General, Visceral, and Thoracic Surgery of the UKE (Hamburg) for the collection of intestinal tissues. This work was supported by the Deutsche Forschungsgemeinschaft (DFG) through the SFB841 and Daisy Huët Röell Foundation.

## Author contributions

M.J.B. and A.F.S. designed the experiments. F.S. and R.R.C.E.S. contributed to experiments; D.P. and K.R. collected the tissue samples; S.L., C.K., and M.A. contributed to the study design, interpretation of the data, and manuscript revision; A.F.S. performed the data analyses. A.F.S. and M.J.B. wrote the manuscript with input from all authors. M.J.B. supervised the study.

## Additional information

**Competing interests:** The authors declare no competing interests.

