## [Peer Review File · Nature Communications]

Reviewers' comments:

Reviewer #1 (NK regulation/development, ILC)(Remarks to the Author):

The authors study innate lymphocyte populations in the human intestine of pediatric patients compared to blood and intestinal samples from adults. Using 18 color flow cytometry, lineage negative lymphocytes are analyzed in the intestinal epithelium and lamina propria compared to blood. Major NK cells subsets (expressing CD56) as well as ILCs (expressing CD127) were observed and further characterized. Several differences were noted between infant and adult samples, including a higher frequency of NK cells in children that translates to a 15-fold increase in these cytotoxic effectors. Several markers of tissue-residency are studied (CD69, CD103, CD49a) that suggest heterogeneity with both tissue-resident and non-resident cells. Using NK maturation markers (KIR, CD94) further heterogeneity was observed with both mature KIR+ NK cells as well as immature NK cells. Transcription factor studies identified a predominant EOMES+ subset in infants that co-express CD103. This subset is rather minor in adult intestine. Concerning NK cell function, cytotoxic effectors Granzyme B and perforin are analyzed; again marked heterogeneity is observed in infant samples that includes cells with apparently higher levels of these proteins and other subsets that resemble that found in adult tissues. While killing assays were not performed due to insufficient cell numbers, cytokine production by infant and adult NK cells appeared relatively similar with no significant differences observed. Finally, the authors observe that T cells are more abundant in adult samples and accordingly, NK/T ratios decrease with increasing age.

This study is well performed and provides a solid characterization of infant NK cell subsets. In contrast, CD127+ ILCs are not studied in any depth. As several of the NK cell observations confirm previous publications, the novel aspects of this manuscript are not so clear to this reader. For example, previous studies provided insights into early emergence of NK cells in humans (Calleja et al 2011; other studies from Roy lab) and a comprehensive paper from the Reiner lab (Collins et al JCI Insight 2017) described EOMES+ NK cells subsets in several fetal tissues with an in-depth characterization of their phenotype and functions. A recent JEM report published an in-depth analysis of NK and ILC in fetal gut (ref. 16).

Another issue that is not adequately addressed concerns the substantial heterogeneity of NK cells and CD127+ ILCs in these samples. The authors identify several NK cell subsets that vary with respect to tissue-residency and other markers, but fail to correlate these subsets with effector functions. In the end, it is not clear how the many changes that occur early in life (microbiota, influx of T cells) affect these different subsets and what is the relevance for intestinal homeostasis.

Reviewer #2 (NK biology during pregnancy/in pediatrics)(Remarks to the Author):

It is very rare to be totally enthusiastic about a submitted paper, but this is the case, which happened very seldom in my life. I am impressed by the amount of work engaged, the controls, and of course the results. The authors describe a very early population of early nk cells in the infant intestine in the very early life as well as its replacement later on by Eomes T cells which colonize the intestine and NK cells declining, with a population of NKp44 CD103 CD69 ieILCs persisting in adult intestines. I say here a novel population because the coupling of phenotypic properties and high lytic activities described are akin to an original subset
this paper is fundamentally and clinically important and should be published asap as such

Reply to reviewers

Regarding manuscript entitled "Tissue-resident Eomes+ NK cells are the major innate lymphocyte population in the human intestine early in life" (NCOMMS-18-22146-T).

Reviewer #1 (NK regulation/development, ILC)(Remarks to the Author):

The authors study innate lymphocyte populations in the human intestine of pediatric patients compared to blood and intestinal samples from adults. Using 18 color flow cytometry, lineage negative lymphocytes are analyzed in the intestinal epithelium and lamina propria compared to blood. Major NK cells subsets (expressing CD56) as well as ILCs (expressing CD127) were observed and further characterized. Several differences were noted between infant and adult samples, including a higher frequency of NK cells in children that translates to a 15-fold increase in these cytotoxic effectors. Several markers of tissue-residency are studied (CD69, CD103, CD49a) that suggest heterogeneity with both tissue-resident and non-resident cells. Using NK maturation markers (KIR, CD94) further heterogeneity was observed with both mature KIR+ NK cells as well as immature NK cells. Transcription factor studies identified a predominant EOMES+ subset in infants that co-express CD103. This subset is rather minor in adult intestine. Concerning NK cell function, cytotoxic effectors Granzyme B and perforin are analyzed; again marked heterogeneity is observed in infant samples that includes cells with apparently higher levels of these proteins and other subsets that resemble that found in adult tissues. While killing assays were not performed due to insufficient cell numbers, cytokine production by infant and adult NK cells appeared relatively similar with no significant differences observed. Finally, the authors observe that T cells are more abundant in adult samples and accordingly, NK/T ratios decrease with increasing age.

This study is well performed and provides a solid characterization of infant NK cell subsets. In contrast, CD127+ ILCs are not studied in any depth.

- We would like to thank the reviewer for the positive overall evaluation of the manuscript, recognizing the in depth analyses of human NK cells in tissues during development. Further, we greatly appreciate the suggestion to assess CD127⁺ ILCs, at present there is a lack of data regarding this population in infant intestinal tissues. The new analyses of CD127⁺ ILCs allowed us to provide additional data regarding this tissue-resident lymphocyte population in infant intestines and their changes during development. Detailed description of the novel CD127⁺ ILCs is described below and shown in **Supplementary 1c-d and Supplementary figure 6a-f**. All changes to the manuscript are highlighted in blue.

As several of the NK cell observations confirm previous publications, the novel aspects of this manuscript are not so clear to this reader. For example, previous studies provided insights into early emergence of NK cells in humans (Calleja et al 2011; other studies from Roy lab) and a comprehensive paper from the Reiner lab (Collins et al JCI Insight 2017) described EOMES+ NK cells subsets in several fetal tissues with an in-depth characterization of their phenotype and functions. A recent JEM report published an in-depth analysis of NK and ILC in fetal gut (ref. 16).

- We thank the reviewer for highlighting these publications describing NK cells during human immune ontogeny and have added these publications to the revised manuscript. Calleja et al. (2011) compared samples of infants and adults with celiac disease, the median age of the children with celiac disease was 4.6 years and 8.2 years for healthy controls. Furthermore, Collins et al. (2017) investigated NK cell development in other fetal tissues, such as liver and lung but not the intestine and described the relative immaturity of NK cells compared to adult blood. The manuscript by Li et al (2018) was included in the previous version of the manuscript and focused on the plasticity between phenotypic NK cells and CD127⁺ ILCs. Taken together, data regarding NK cell dynamics in the intestine in the first year of life of children without an inflammatory or autoimmune disease are currently lacking due to the difficulties to obtain intestinal samples of this population. The samples collected in our study

provided the opportunity to investigate NK cells dynamics during this critical phase of human development and demonstrated for the first time that NK cells are the dominant innate lymphocyte early in life after birth and subsequently decrease, while NKp44⁺CD69⁺ cells persist in adult intestines. We have now included these references (ref #20, ref #48), highlighted by the reviewer in the revised manuscript, as suggested.

Another issue that is not adequately addressed concerns the substantial heterogeneity of NK cells and CD127⁺ ILCs in these samples.

- We thank the reviewer for the suggestion to perform analyses of CD127⁺ ILCs and highlighting the heterogeneity of NK cells and CD127⁺ ILCs related to specific molecules associated with tissue-residency. These new analyses have allowed us to add novel data regarding CD127⁺ ILCs dynamics early in life. Next to NK cells also CD127⁺ ILCs were more frequent early in life (**Supplemental figure 1c-d**), indicating that infant intestines harbored several innate lymphocyte populations with larger frequencies compared to adult intestines. Infant lamina propria-derived CD127⁺ ILCs showed differential expression of CD103 and CD49a compared adult lamina propria-derived CD127⁺ ILCs (**Supplemental figure 6a**). Infant and adult epithelial and lamina propria-derived CD127⁺ ILCs had a high expression of CD69, whereas classical NK cell features such as Eomes, perforin and granzym B were low (**Supplemental figure 6b-f**), confirming a non-cytotoxic phenotype of this CD127⁺ ILC population. The new analyses furthermore showed for the first time that frequencies and absolute numbers of both epithelial and lamina propria-derived CD127⁺ ILCs rapidly decreased directly after birth (**Supplemental figure 1c-d**). Thus although there is plasticity between NK cells and CD127⁺ ILCs, NK cells are not decreasing due to their development into CD127⁺ ILCs. We thank the reviewer for the suggestion to incorporate these new analyses showing that early in life a more heterogeneous innate lymphocyte population including of NK cells and CD127⁺ ILCs exists in infant intestines. These new data are now included and discussed in the revised manuscript.

The authors identify several NK cell subsets that vary with respect to tissue-residency and other markers, but fail to correlate these subsets with effector functions.

- The reviewer makes an important point to further strengthen the manuscript, by performing a functional assessment of NK cells expressing different molecules for residency. In the overall analyses CD107 was differentially expressed between infant and adult NK cells (**Fig. 6d**), thus next we assessed CD107 expression upon PMA stimulation on CD103, CD49a or CD69⁺ NK cell subsets. Similarly to the analyses of surface markers and transcription factors, CD69⁺NK cells performed differently compared to CD103⁺ NK cells, with CD69⁺NK cells having lower CD107-expression than CD103⁺ NK cells. This difference was particularly strong in adult cells (**figure 6e**). These new data further supporting a differential phenotype of CD69⁺ NK cells and are now added to the revised manuscript.

In the end, it is not clear how the many changes that occur early in life (microbiota, influx of T cells) affect these different subsets and what is the relevance for intestinal homeostasis.

- Immunity in early life is faced with extraordinary challenges due to the transitioning from the protected in utero environment to the adaptation to microbial invasion, introduction of nutrient-antigens and unmatched tissue-growth. All of these aspects are important actors in the modulation of immune responses. Mouse studies provide the opportunity to investigate these complex interactions systematically, however are limited due to the critical differences between mice and humans. Human tissue-based studies bear the important limitation that longitudinal samples before and after an intervention, are practically impossible. Assessment thus far of the developing immune system in the first year of life therefore still largely relies on blood samples, which is a poor reflection of immunity in the tissues. Our study aims to address these points by investigating in a unique cross-sectional cohort including individuals ranging from newborn infants to adults without an inflammatory disease to determine

these dynamic changes during immune ontogeny in intestines. The analyses of human samples demonstrate a consistent pattern of maturation of innate lymphocyte compartment in intestines. The limitations raised by the reviewer are now discussed in the revised version of the manuscript.

Reviewer #2 (NK biology during pregnancy/in pediatrics) (Remarks to the Author):

It is very rare to be totally enthusiastic about a submitted paper, but this is the case, which happened very seldom in my life. I am impressed by the amount of work engaged, the controls, and of course the results. The authors describe a very early population of early nk cells in the infant intestine in the very early life as well as its replacement later on by Eomes T cells which colonize the intestine and NK cells declining, with a population of NKp44 CD103 CD69 ielLCs persisting in adult intestines. I say here a novel population because the coupling of phenotypic properties and high lytic activities described are akin to an original subset this paper is fundamentally and clinically important and should be published asap as such

- We would like to thank the reviewer for the very positive assessment of the work and results described in the manuscript.

REVIEWERS' COMMENTS:

Reviewer #1 (Remarks to the Author):

The authors have included additional information concerning CD127+ ILCs in the revised manuscript. While the phenotypic analysis of gut ILCs is rather expected (tissue resident lacking cytotoxic effectors), the inclusion of other functional outputs (cytokine signatures such as IL-5, IL-13, IL-17A, IL-22) would have added substantial new information. Do the authors have this data?

The revised manuscript now adequately references the literature.

Reply to Reviewer #1 :

We much appreciate the comment of the reviewer to include additional cytokines produced by ILCs in the analyses. However, the flow cytometric panels did not include these cytokines, therefore, we are unable to the add these data to the revised manuscript. We have included the suggestion in the discussion for future studies.